# Influence of Ultra-Processed Foods Consumption on Redox Status and Inflammatory Signaling in Young Celiac Patients

**DOI:** 10.3390/nu13010156

**Published:** 2021-01-06

**Authors:** Teresa Nestares, Rafael Martín-Masot, Marta Flor-Alemany, Antonela Bonavita, José Maldonado, Virginia A. Aparicio

**Affiliations:** 1Department of Physiology, Faculty of Pharmacy, University of Granada, 18071 Granada, Spain; nestares@ugr.es (T.N.); floralemany@ugr.es (M.F.-A.); antonelabonavita@hotmail.com (A.B.); 2Institute of Nutrition and Food Technology “José Mataix Verdú” (INYTA), Biomedical Research Centre (CIBM), University of Granada, 18100 Armilla, Spain; 3Pediatric Gastroenterology and Nutrition Unit, Hospital Regional Universitario de Malaga, 19010 Málaga, Spain; rafammgr@gmail.com; 4Sport and Health University Research Centre (iMUDS), University of Granada, 18100 Armilla, Spain; 5Department of Pediatrics, University of Granada, 18071 Granada, Spain; jmaldon@ugr.es; 6Biohealth Research Institute, 18071 Granada, Spain; 7Maternal and Child Health Network, Carlos III Health Institute, 28029 Madrid, Spain; 8Pediatric Clinical Management Unit, “Virgen de las Nieves” University Hospital, 18071 Granada, Spain

**Keywords:** celiac disease, ultra-processed foods, gluten-free diet, inflammatory signaling, oxidative stress, children

## Abstract

The current study was designed to assess the influence of consumption of ultra-processed (UPF) on oxidative/antioxidant balance and evoked inflammatory signaling in young patients with celiac disease (CD). The study included 85 children. The celiac group (*n* = 53) included children with CD with a long (>18 months, *n* = 17) or recent (<18 months, *n* = 36) adherence to a gluten-free diet (GFD). The control group (*n* = 32) included healthy children with a significantly lower consumption of UPF compared to the CD group, both expressed as kcal/day (*p* = 0.043) and as percentage of daily energy intake (*p* = 0.023). Among children with CD, the group with the lowest consumption of UPF (below the 50% of daily energy intake) had a greater Mediterranean diet (MD) adherence and higher moderate physical activity levels. In addition, CD children with the lowest consumption of UPF had healthier redox (lower soluble superoxide dismutase-1 and 15-F2t-isoprostanes) and inflammatory profiles (lower macrophage inflammatory protein-1α) compared to the group with the highest consumption of UPF (all, *p* < 0.05) regardless of the time on a GFD. These findings highlight the importance of a correct monitoring of the GFD. An unbalanced GFD with high consumption of UPF and an unhealthy pattern with less physical activity and worse adherence to MD results in a worse inflammatory profile, which could act as a parallel pathway that could have important consequences on the pathophysiology of the disease.

## 1. Introduction

Celiac disease (CD) occurs in about 1–1.4% of people in most populations [1]. It is a multifactorial, autoimmune disorder, caused by an immune reaction which is triggered by the ingestion of gluten and related proteins. This occurs in people who carry the DQ2 and/or DQ8 Human Leukocyte Antigen (HLA) class II haplotypes. Aa variable combination of high CD-specific antibody titers, an inflammatory enteropathy with different degrees of severity and a wide range of digestive and/or systemic symptoms [2] characterize this disease. 

The pathogenesis of CD is complicated and still not fully explained. Gluten causes an abnormal innate and adaptive immune response in patients, generating autoantibodies [3] that can affect not only the intestine but the entire organism. Oxidative stress has an important function in the pathogenesis of many intestinal manifestations of CD; the main processes possibly involved in gliadin toxicity are a higher concentration of reactive oxygen species (ROS), a lower antioxidant capacity and a pro-inflammatory state [4]. Gliadin gene sequence contains regions responsible for triggering oxidative stress and inducing the release of proinflammatory cytokines like interleukin 1β (IL1β) [5] and interleukin 15 (IL-15) [6]. Due to an adaptive immune response, various epitopes of gliadin can stimulate tumor necrosis factor (TNF) and other proinflammatory cytokines. Th1 response increases interferon gamma (IFNγ), that leads to intraepithelial lymphocyte toxicity and onset of CD [7]. On the other hand, gut microbiota seems to play an important role in the pathogenesis of the disease, and changes in its composition have been observed in celiac children [8].

The only available treatment for CD is a lifelong strict gluten-free diet (GFD) is. The improvement and resolution of symptoms typically occur within days or weeks and often precedes the normalization of serological markers and duodenal villous atrophy [9]. A GFD in children must be suitable for growth and pubertal development [10]. However, the nutritional adequacy of the GFD remains controversial and some evidence suggests that GFD is nutritionally unbalanced [11,12,13]. The scientific literature raises questions about the food choices of individuals with CD because it is a restrictive diet which substitutes naturally gluten-containing products with their ultra-processed gluten-free analogs [14].

Ultraprocessed foods (UPF) are products of food technology, formulated from 5 or more industrial ingredients and containing little or no whole food [15]. They are considered harmful to human health due to their high caloric content (up to 500 kcal/100 g) and low nutritional quality, high glycemic load, because of being rich in sodium, simple sugars and saturated and trans fats, low in fiber, proteins and various micronutrients, and because they contain a large number of additives in order to resemble natural foods as closely as possible [14,16,17,18]. Its production and consumption have increased throughout the world [14]. They are appetizing which makes them able to displace more nutritionally interesting foods and to interfere with the ability to control eating habits [18].

Diverse studies correlated the consumption of UPF with the increase in chronic non-communicable diseases in children and adults [14,16,19,20,21]. However, there are no studies that associate the consumption of UPF with oxidative stress and evoked inflammatory signaling in celiac children in a GFD. Nonetheless, lifestyle changes and including exercise might help in the regulation of redox state and decrease inflammation in children [22,23].

Until now, most studies on the nutritional value of GFD have focused on studying its deficiencies, and an increase in body weight has been described as a consequence of excessive consumption of dietary products rich in vegetable fats (e.g., rapeseed, palm and coconut oil) [24]. However, there have been no large case-control studies on the effect of UPF consumption in celiac children. Therefore, the aim of the present study was to evaluate the influence of UPF consumption on oxidative stress, antioxidant capacity and inflammatory signaling in celiac children on a GFD.

## 2. Materials and Methods

### 2.1. Subjects

The research was carried out following the principles of the Declaration of Helsinki and its later amendments and it was approved by the Ethics Committee of the University of Granada (Ref. 201202400000697). The study included 85 children aged 7–18 years old, who attended the Gastroenterology, Hepatology and Child Nutrition Service from the “Virgen de las Nieves” University Hospital in Granada, Spain.

32 healthy children were included in the control group. Their serological screening was negative and they had no history of any chronic disease. These children attended this Service due to minor symptoms related to chronic functional constipation, according to the Rome IV criteria [25]. They were included in the control group once it was verified that it was due to transitory gastrointestinal symptoms (functional constipation). The inclusion criteria for the control group were the following: (a) age between 7 and 18 years, (b) absence of serum IgA and IgG anti-transglutaminase (tTG) antibodies, (c) normal weight for the age, (d) absence of gastrointestinal disorders in the previous year and (e) normal appetite. Children with diagnosed CD according to the European Society for Pediatric Gastroenterology Hepatology and Nutrition (ESPGHAN) [2] were included in the CD group (*n* = 53). The exclusion criteria for both groups were acute and chronic inflammation, liver or kidney diseases, diabetes, chronic asthma, inflammatory bowel disease, and consumption of dietary supplements that contain substances with antioxidant activity. Obese patients were also excluded (following the criteria of the International Task Force) [26] as well as those who did not sign the informed consent. Written informed consent was obtained from all parents.

### 2.2. Clinical and Socio-Demographics

The same group of researchers assessed participants’ clinical and socio-demographic characteristics (i.e., age, household composition, parents’ marital status, educational level and smoking habit).

The International Physical Activity Questionnaire [27] was used to register physical activity.

### 2.3. Anthropometric Measures

In both the control and the CD groups, Anthropometric characteristics (weight, height) were assessed. A stadiometer (Seca200, Hamburg, Germany) was used to measure height to the nearest 5 mm. The same mechanical balance was used to measure body weight (Seca200, Hamburg, Germany).

### 2.4. Blood Sampling

Venous blood samples from fasting patients were collected into anticoagulated tubes with sodium heparin during morning hours. Blood samples were centrifuged at 2500× *g* at 4 °C for 10 min to obtain plasma. Plasma samples remained frozen (−80 °C) until measurements.

### 2.5. Soluble Superoxide Dismutase (SOD) 1

The HND3MAG-39K Milliplex MAP Human Neurological Disorders Magnetic Bead Panel 3 (Millipore Corporation, Missouri, Saint Louis, MO, USA), based on immunoassays on the surface of fluorescent-coded beads (microspheres), was used to determine Soluble isoforms of SOD1 in plasma following the specifications of the manufacturer (50 events per bead, 50 µL sample, gate settings: 8000–15,000, time out 60 s). A LABScan 100 analyzer (Luminex Corporation, Texas, Austin, TX, USA) with a xPONENT software for data acquisition was used to reads the plate. The average values for each set of duplicate samples or standards were within 15% of the mean. Standard curve: SOD1: 0.04–30 ng/mL. Soluble enzyme concentrations in plasma samples were determined by comparing the mean of duplicate samples with the standard curve for each assay.

### 2.6. 15-F2t-Isoprostanes

The isoprostanes are prostaglandin-like compounds formed in vivo from the free radical-catalyzed peroxidation of essential fatty acids. To measure the isoprostanes in urine, a commercial kit Enzyme Immunoassay for Urinary Isoprostane (Oxford Biomedical Research, Oxford, UK) was used, which is a competitive enzyme-linked immunoassay (ELISA) for determining levels of 15-F2t-Isoprostane (the best characterized isoprostane) in urine samples. In order to eliminate interference due to non-specific binding, the urine samples were mixed with an enhanced dilution buffer. The 15-F2t-Isoprostane in the samples or standards competes with the 15-F2t-Isoprostane conjugated to horseradish peroxidase (HRP) for binding to a polyclonal antibody specific for 15-F2t-Isoprostane coated on the microplate. When the substrate is added, the HRP activity results in color development; the intensity of the color is proportional to the amount of 15-F2t-Isoprostane-HRP bound and inversely proportional to the amount of unconjugated 15-F2t-Isoprostane in the samples or standards. The plate was read spectrophotometrically (Bio-tek, Vermont, Winooski, VT, USA) at 450 nm.

### 2.7. Total Antioxidant Status (TAS)

Peripheral blood was placed in pre-cooled test tubes on the examination day in order to determine plasma TAS levels. The plasma was immediately separated in a refrigerated centrifuge, aliquoted and stored at −20 °C until further use. A TAS Randox kit (Randox Laboratories, Ltd., Crumlin, UK) was used to analyze freshly thawed batches of plasma. Results were expressed in mM of Trolox equivalents. The reference range for human blood plasma is given as 1.30–1.77 mmol/L by the manufacturer. The linearity of calibration extends to 2.5 mmol/L of Trolox. We used measurements in duplicate in order to determine intra-assay variability.

### 2.8. Inflammatory Parameters

The HCYTMAG-60K-PX29 Milliplex MAP Human Cytokine/Chemokine Magnetic Bead Panel (Millipore Corporation, Missouri, Saint Louis, MO, USA) was used following the specifications of the manufacturer to determine in plasma IFN-γ, Interleukin (IL)-10, IL-12P40, IL-12P70, IL-13, IL-15, IL-17A, IL-1α, IL-1β, IL-2, IL-3, IL-4, IL-5, IL-6, IL-7, IL-8, Interferon-inducible protein (IP)-10, Monocyte chemoattractant protein (MCP)-1, Macrophage inflammatory protein (MIP)-1α, MIP-1β, TNF-α, TNF-β and Vascular endothelial growth factor (VEGF). The plate was read on a LABScan 100 analyzer (Luminex Corporation, Texas, Austin, TX, USA) with a xPONENT software for data acquisition. Standard curve: 3.2–10.000 pg/mL. Cytokines concentrations in plasma samples were determined by comparing the mean of duplicate samples with the standard curve for each assay.

### 2.9. Dietary Assessment

A three-day food record, two on weekdays and one on the weekend was used to assess the dietary intake. The same trained nutritionist carefully explained face to face the diary to the children and their parents and accompanied with a photographic atlas including different portion-size food pictures and a set of household measures and detailed instructions for its compilation [28]. All foods consumed during the different meals (breakfast, morning snack, lunch, afternoon snack, dinner) were included in the survey. For each meal, participants were requested to report an exhaustive description of the food and the recipes (including cooking methods and sugar or fats added during the meal preparation), food amount (according to the atlas) and the brands of packaged foods consumed. 

Then foods were classified according to the NOVA classification, which has been employed in many studies conducted in several countries and is the most frequently used method to examine diets according to food processing, [17]. International bodies including PAHO, WHO and FAO also recognize and use it [18,29]. NOVA is a food classification based on the extent and purpose of industrial food processing, which classifies foods into four groups: unprocessed and minimally processed foods, processed culinary ingredients, processed foods and ultra-processed foods [30]. The latter category encompasses a group of industrial formulations that are manufactured using several ingredients and a series of processes. In order to classify the foods as unprocessed or minimally processed, culinary ingredients, processed and ultra-processed foods based on the NOVA classification, the three-day food records were analyzed [30]. The total dietary energy intake was calculated for each individual. Subsequently, the energy (kcal/day) and percentage of calories (% of the total daily energy intake) derived from each category of the NOVA classification food item was calculated. The same trained nutritionist analyzed all diaries using the *Evalfinut* software that includes the Spanish Food Composition Database [31]. The specific composition of gluten and gluten-free UPF (from labels) were introduced in the software when calculating dietary balance. The mean of the three-day food records was employed in the present analyses.

Moreover, the Mediterranean diet Quality Index in Children and Adolescents (KIDMED) survey was employed to assess adherence to the Mediterranean diet (MD) [32]. The KIDMED index ranges from 4 (no adherence to the MD) to 12 (complete adherence to the MD). This index was determined using a 16-point questionnaire, which assesses various dietary habits. Each answer was scored according to its consistency with habits associated with the Mediterranean dietary pattern. Then the scores were added up to quantify the total index of the participant’s adherence to the MD. 

### 2.10. Data Analyses 

Descriptive statistics (mean standard deviation) for quantitative variables and percentage of participants (%) for categorical variables were employed to describe the baseline characteristics of the study sample. To explore differences in the continuous variables, the Student-*t* test was conducted. Furthermore, differences in categorical variables were assessed by using the Chi-squared test. 

We employed a one-way analysis of covariance (ANCOVA) after adjustment for age, sex and following a GFD for at least 18 months to assess differences in food groups consumption and NOVA food classification of children by CD (celiac group vs healthy group). The MD adherence by percentage of energy consumed from UPF in celiac children (below 50% of daily energy intake vs. above 50% of daily energy intake) was compared by ANCOVA after adjusting for age, sex and following a GFD for at least 18 months. Moderate physical activity (METS/week) by percentage of energy consumed from UPF in celiac children (below 50% vs. above 50% of daily energy intake) was compared by ANCOVA after adjusting for following a GFD for at least 18 months. An ANCOVA after adjusting for moderate physical activity, following a GFD for at least 18 months and the MD adherence was also employed to assess differences in levels of oxidative/antioxidant biomarkers and inflammatory profiles between celiac children consuming UPF below 50% of daily energy, above 50% of daily energy and control children. Post-hoc multiple comparisons (Bonferroni’s correction) were applied to examine pairwise differences between groups (e.g., celiac disease with daily energy intake from UPF below 50% vs. controls). In those outcomes where outliers were/remained influential, we employed a subtle variation of winsorizing (convert back from a z-score: replacing extreme scores (z > 2.58) with a score equivalent to ±2.58 standard deviations from the mean) in order to handle these outliers.

Differences in dietary habits in CD children by percentage of energy intake from ultra-processed foods (below 50% vs. above 50%) were assessed with an ANCOVA after adjusting for following a GFD for at least 18 months (Appendix A). 

We conducted the data analyses with the Statistical Package for Social Sciences (IBM SPSS Statistics for Windows, Version 22. IBM Corp.: Armonk, NY, USA), and the statistical significance was set at *p* ≤ 0.05.

## 3. Results

The baseline sociodemographic, dietary and anthropometric characteristics and physical activity levels of the study sample are shown in Table 1. A total of 53 children with CD participated in the study (mean age 9.6 ± 3.8 years). More than half of the participants with CD followed a GFD (68%) for less than 18 months and had a medium-high MD adherence (88%). The control group included 32 healthy children (mean age 10.7 ± 4.0 years). There were differences in height (*p* = 0.048) and weight (*p* = 0.018) between groups.

Differences in food groups consumption and NOVA food classification of children by CD (celiac vs. control group) are shown in Table 2. The celiac group had a significantly higher consumption of UPF compared to the control group, both expressed as kcal/day (*p* = 0.043) and as percentage of daily energy intake (*p* = 0.023). After further adjusting the model for following a GFD for at least 18 months, there were no statistically significant differences but CD children showed a trend close to signification indicating a higher percentage of energy intake from UPF among celiac compared to controls (*p* = 0.056). 

The Mediterranean diet adherence and moderate physical activity levels by percentage of daily energy consumed from UPF in celiac children is shown in Figure 1. The group with the highest intake of energy from UPF (above the 50% of total energy) showed a lower MD adherence than the group with the lowest intake (below the 50% of total energy) in both, the unadjusted model (*p* = 0.041) and after adjusting the model for age, sex and following a GFD for at least 18 months (*p* = 0.046). The group with the highest intake of energy from UPF (above the 50% of total energy) showed lower levels of moderate physical activity than the group with the lowest intake (below the 50% of total energy) in both, the unadjusted model (*p* = 0.037) and after adjusting for following a GFD for at least 18 months (*p* = 0.040).

Differences in levels of oxidative/antioxidant biomarkers and inflammatory profiles between CD children by percentage of energy from UPF (below 50% vs. above 50%) and the control group are shown in Table 3. After adjusting the model for physical activity levels, following a GFD for at least 18 months and MD adherence, levels of SOD1 (oxidative stress-mediated antioxidant), 15-F2t-isoprostanes (biomarker related to damage to the prostaglandins) and MIP-1α (pro-inflammatory cytokines) were lower in CD children consuming UPF below 50% of daily energy intake compared to CD children consuming UPF above 50% of daily energy intake (all, *p* < 0.05). In addition, CD children consuming UPF above 50% of daily energy had higher levels of SOD1, IFN-γ and MIP-1α compared to healthy controls (all, *p* < 0.05).

Differences in dietary habits in CD children by percentage of energy intake from UPF (below 50% vs. above 50%) are shown in Appendix A. After adjusting for following a GFD for at least 18 months, CD children with more than 50% of daily energy intake from UPF showed a lower intake of whole dairy products (*p* = 0.006) and a trend to higher intake of poultry (*p* = 0.076) and a lower intake of vegetables (*p* = 0.077).

## 4. Discussion

Our results suggest that the consumption of UPF was higher in the CD children compared to the control group. Among CD children, those with a lower intake of UPF showed better inflammatory signaling and oxidative status (i.e., lower values for oxidative biomarkers and pro-inflammatory cytokines).

The only effective treatment for CD is a life-long strict GFD which, apart from maintaining the safe limit of gluten intake, must also be nutritionally balanced to ensure a healthy life. However, a GFD is a restrictive diet whose nutritional adequacy remains controversial with some evidence suggesting that GFD is nutritionally unbalanced [11,12,13]. The scientific literature raises questions concerning the food choices of individuals with CD because they substitute natural gluten-containing products with their ultra-processed gluten-free analogs [14]. This could have a negative effect on health, and it should be seriously taken into account, since the limited choice of food products in the diet of children with CD could induce a high consumption of UPF, such as snacks and biscuits. Our results showed that, in general, CD children had a greater percentage of daily energy intake from UPF and drink products than the control children. This could be partially explained because, within the range of gluten-free foods, CD children usually choose those that are made gluten free through a process of purification instead of those that are naturally gluten free, leading to a diet higher in fat and carbohydrates concentration [12,13]. However, after adjusting UPF consumption for time of GFD (18 months), we observed that the differences disappeared, which may be due to a better adherence to the GFD and a better balance of macro and micronutrients among these patients, which highlights the need for close monitoring diet quality after the establishment of a GFD [33].

The consumption of UPF has been directly related to increased mortality and the appearance of chronic non-communicable diseases, especially cardiovascular diseases, but also obesity, cancer or depression in the adult population [34]. There are several mechanisms which might explain this relationship including a lower consumption of vitamins A, B_12_, C, E, calcium, zinc, fiber and polyunsaturated fatty acids, and a higher intake of trans fats, sodium and sugars [20,21,35,36,37,38,39]. Furthermore, this eating pattern negatively affect the gut microbiota through the appearance of intestinal dysbiosis, which can trigger a pro-inflammatory immune response and an increase in intestinal permeability [14].

Despite the fact that there are numerous studies showing the defects of GFD, only a few have assessed its effect on the clinical course and there are hardly any studies that assess its role in the physiopathology of the disease. It is known that gut microbiota disturbance [40] plays a key role in the pathogenesis of CD, as colonizing gut bacteria are critical for the normal development of host defense [41] for its metabolic and protective function of the host [42]. Various studies have revealed an alteration in the microbiota of celiac patients compared to the healthy controls, observing a reduction in the population of *Lactobacillus* and *Bifidobacterium* and an increase especially in *Bacteroides*, *Escherichia coli* and *Clostridium leptum* [8,43]. Furthermore, a profile of bacterial proteases, capable of hydrolyzing gliadin, has been described in celiac patients that was absent in healthy ones, conforming a different bacterial proteolytic activity [44]. In this sense, the fact that the CD children in our study had a higher intake of UPF may aggravate the pathophysiology of the disease. On the other hand, the fact that CD children consume more UPF could be because of, but not only, the characteristics of a poorly completed GFD and, although less likely, because of a less healthy consumption pattern in this group prior to the onset of the disease, due to the relationship of UPF with the development of intestinal pathologies [14].

Contrarily, unprocessed and minimally processed food-based diets have shown the capacity to promote gut microbiota eubiosis, anti-inflammatory response, and epithelial integrity through bacterial butyrate production [45]. Indeed, a higher consumption of UPF by CD children in our study leads to a less favorable redox and inflammatory profile (compared to CD children consuming less than 50% of daily energy intake from UPF and controls), which altogether could also aggravate the pathophysiology of their disease. It is known that the oxidative status present in certain chronic diseases such as CD, especially at diagnosis, is caused by the interaction of gluten in the lamina propria, acting at the local and genomic level with an intracellular oxidative imbalance, characterized by increased levels of lipid peroxidation products [46,47] which can induce the formation of oxidative DNA lesion products (8-OHdG) [48]. Additionally, CD patients present a severe reduction of antioxidant capacity (including antioxidant vitamins) [49]. It is known that ROS signaling can enhance the synthesis of inflammatory mediators such as TNF-α and IL-1 [50]. In fact, we previously [4] reported the increase in these inflammatory molecules in children with CD who followed a GFD. Interestingly, these changes are maintained after adjusting for GFD time, which could be due to a strong enough response to induce an inflammatory state which is maintained and probably a parallel pathway in the pathogenesis of chronic inflammation and the oxidative status of uncontrolled disease. In this sense, we can assume that there is another source of the deleterious pro-inflammatory cytokines. Present results suggest that a higher UPF consumption and lower physical activity levels could be responsible. A healthy lifestyle includes meeting the recommendations regarding minimum levels of physical activity. It is well established that adequate physical activity levels promote better inflammatory and redox profiles in the general adult population [51,52]. Notwithstanding, there are specific ways in which the balance between pro- and anti-inflammatory, oxidant-antioxidant factors associated with exercise can influence health and growth in children [22]. Therefore, the influence of physical activity and exercise on inflammatory signal and redox status in children remains unclear. In fact, SOD concentrations were lower among our CD children with UPF below 50% of daily energy intake, which may suggest a lower need of release of this relevant antioxidant enzyme in this group of patients. It should be taken in account that SOD levels usually increase with exercise among adult populations [52]. However, oxidative stress responses to exercise and the underlying mechanisms in the pediatric population are still unclear [53]. Therefore, not only diet matters, and it might be mandatory that CD children exercise for a better prognosis of the disease [23]. Studies exploring the influence of objectively measured physical activity levels and different exercise programs on CD patients are warranted in order to better design effective lifestyle interventions.

In our study, the adherence to the MD was inversely related to the intake of UPF, in a similar way to that described previously [54]. Notwithstanding, small differences were found regarding dietary habits between CD children according to UPF intake. In fact, CD children with more than 50% of daily energy intake from ultra-processed foods showed a lower intake of whole dairy products and a trend to higher intake of poultry and lower of vegetables. As a result, although a tendency towards healthier habits has been observed in CD children with less than 50% of daily energy intake from UPF, the differences might not be strong enough to reflect changes in the overall dietary pattern. Consequently, the global index of adherence to the MD may not be sensitive enough. Indeed, even after adjusting our results for MD adherence, patients who consumed less than 50% of their daily energy intake from UPF presented a more favorable oxidative-antioxidant and inflammatory profiles than those with greater UPF consumption. The MD is characterized by being a diet with high antioxidant power and with nutrigenomic modulation capacity, showing itself as a protective factor against various diseases [55], but in recent years there has been a greater consumption of processed and ultra-processed foods of animal origin in the pediatric population [56]. In this sense, other studies have correlated the consumption of UPF and the development of diseases adjusting for adherence to the MD and they failed to confirm the present results [57]. 

## 5. Strengths and Limitations

There are some limitations that must be underlined. Firstly, the present study sample size was relatively small and it should be noted that since it is a cross-sectional study we cannot establish causal relationships, and consequently the present results must be interpreted with caution. Secondly, the assessment of physical activity levels was self-reported (through the IPAQ) instead of objectively measured through accelerometry, which constitute the gold standard. Based on the present findings, it is advisable to recruit a higher number of participants to confirm or contrast the present findings.

## 6. Conclusions

Overall, our results indicate that CD children who consumed more UPF and performed less physical activity presented higher levels of oxidative stress and some pro-inflammatory cytokines, regardless of the time on a GFD. This suggests that for a better redox and inflammatory profile it is necessary to promote healthy lifestyle habits which include improvements in the quality of the diet, regardless of CD. An unhealthy dietary pattern it is shown as a parallel pathway that could have important consequences in terms of cellular aging and long-term prognosis in CD children. Among the healthy living patterns, the adherence to a MD might be a key factor, which was inversely associated with the consumption of UPF.

## Figures and Tables

**Figure 1 nutrients-13-00156-f001:**
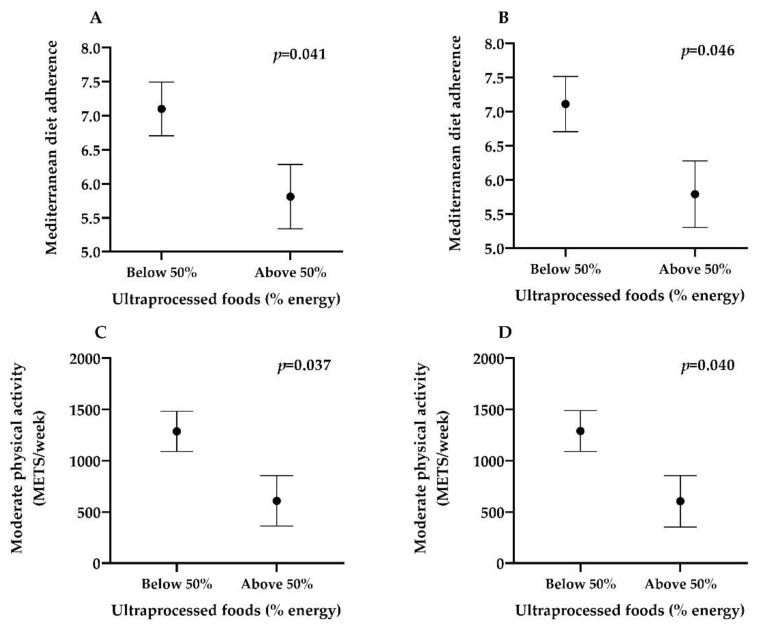
Mediterranean diet adherence and moderate physical activity levels (METS/week) by percentage of energy consumed from ultra-processed foods. Dots represent mean and bars represent standard error. (**A**) Mediterranean diet adherence by energy intake from ultra-processed foods below 50% vs. energy intake from ultra-processed foods above 50%. Model unadjusted. (**B**) Mediterranean diet adherence by energy intake from ultra-processed foods below 50% vs. energy intake from ultra-processed foods above 50%. Model adjusted for age, sex and following a gluten free diet for at least 18 months. (**C**) Moderate physical activity (METS/week) by energy intake from ultra-processed foods below 50% vs. energy intake from ultra-processed foods above 50%. Model unadjusted. (**D**) Moderate physical activity (METS/week) by energy intake from ultra-processed foods below 50% vs. energy intake from ultra-processed foods above 50%. Model adjusted for following a gluten free diet for at least 18 months.

**Table 1 nutrients-13-00156-t001:** Sociodemographic characteristics of the study participants.

Variable	Celiac Group (*n* = 53)	Control Group (*n* = 32)	*p*
Age (years)	9.6 (3.8)	10.7 (4.0)	0.187
Sex (female, n [%])	37 (69.8)	13 (40.6)	0.008
Weight (kg)	32.9 (12.3)	40.1 (15.0)	0.018
Height (cm)	134.6 (21.1)	144.3 (22.7)	0.048
Physical activity levels (METS/week)			
Moderate physical activity (*n* = 81)	981 (1061)	937 (825)	0.842
Vigorous physical activity (*n* = 81)	1425 (1548)	1599 (1644)	0.632
Mediterranean diet adherence n (%) (*n* = 81)			
Low	6 (11.8)	1 (3.1)	0.199
Medium	23 (45.1)	19 (63.3)	
High	22 (43.1)	11 (33.3)	
Following a gluten-free diet for at least 18 months n (%)			
Yes	17 (32.1)	-	
No	36 (67.9)	-	
Parents’ marital status (% married) (*n* = 68)	39 (97.5)	28 (100)	0.339

SD, standard deviation; Values shown as mean ± SD unless otherwise indicated. METS, Metabolic equivalents.

**Table 2 nutrients-13-00156-t002:** Differences in food groups consumption and NOVA food classification of children by celiac disease (celiac vs. control group).

	Celiac Group (*n* = 53)	Control Group (*n* = 32)	*p* ^a^	*p* ^b^
**Energy (kcal/day)**	1905 (69.4)	1839 (90.4)	0.571	0.818
**NOVA food classification**				
Unprocessed or minimally processed foods (kcal/day)	654 (42.0)	733 (55.6)	0.272	0.325
Unprocessed or minimally processed foods (%E)	35.5 (1.9)	40.6 (2.6)	0.129	0.273
Processed culinary ingredients (kcal/day)	113 (12.8)	98 (16.7)	0.490	0.289
Processed Foods (kcal/day)	213 (28.8)	278 (37.5)	0.182	0.117
Ultra-processed food and drink products (kcal/day)	920 (54.9)	730 (71.6)	0.043	0.119
Ultra-processed food and drink products (%E)	47.0 (2.2)	38.6 (2.8)	0.023	0.056

E, energy. ^a^ Model adjusted for age and sex. ^b^ Model additionally adjusted for following a gluten free diet for at least 18 months. Values shown as mean (standard error).

**Table 3 nutrients-13-00156-t003:** Differences in oxidative/antioxidant biomarkers and inflammatory profiles in celiac children by percentage of energy intake from ultra-processed foods (below 50% vs. above 50%) and control children.

	Celiac Children Below 50%	Celiac Children Above 50%	Control Children	*p* ^a^	*p* ^b^
**Oxidative/antioxidant biomarkers**					
SOD1 (pg/mL)	87.7 (13.4) (*n* = 16) ^a^	148.2 (16.6) (*n* = 13) ^a,b^	83.7 (12.6) (*n* = 18) ^b^	0.014	0.020
15-F2t-isoprostanes (pg/mL)	8.3 (0.3) (*n* = 22) ^a^	9.9 (0.4) (*n* = 18) ^a^	9.2 (0.3) (*n* = 27)	0.008	0.004
TAS (mmol/L)	1.7 (0.1) (*n* = 22)	1.5 (0.1) (*n* = 18)	1.6 (0.1) (*n* = 26)	0.154	0.144
**Inflammatory markers**					
IFN-γ (pg/mL)	45.8 (8.1) (*n* = 22)	69.6 (9.2) (*n* = 15) ^a^	38.8 (7.6) (*n* = 26) ^a^	0.043	0.047
IL-10 (pg/mL)	11.8 (2.0) (*n* = 22)	16.5 (2.3) (*n* = 18)	12.3 (1.9) (*n* = 26)	0.265	0.239
IL-12P40 (pg/mL)	32.9 (5.2) (*n* = 16)	44.3 (6.1) (*n* = 12)	40.6 (4.9) (*n* = 19)	0.331	0.478
IL-12P70 (pg/mL)	8.5 (1.1) (*n* = 22)	10.5 (1.3) (*n* = 18)	8.8 (1.0) (*n* = 26)	0.456	0.415
IL-13 (pg/mL)	39.5 (18.1) (*n* = 16)	54.2 (22.4) (*n* = 10)	52.7 (18.1) (*n* = 16)	0.842	0.953
IL-15 (pg/mL)	4.9 (0.9) (*n* = 17)	6.5 (1.0) (*n* = 14)	5.9 (0.9) (*n* = 21)	0.529	0.721
IL-17A (pg/mL)	6.9 (1.1) (*n* = 20)	8.6 (1.2) (*n* = 17)	6.3 (1.0) (*n* = 24)	0.334	0.364
IL-1α (pg/mL)	33.5 (5.9) (*n* = 21)	43.9 (6.8) (*n* = 17)	32.5 (5.6) (*n* = 24)	0.376	0.425
IL-1β (pg/mL)	4.6 (0.5) (*n* = 22)	4.4 (0.6) (*n* = 18)	4.8 (0.5) (*n* = 26)	0.903	0.913
IL-2 (pg/mL)	3.4 (0.4) (*n* = 21)	3.5 (0.5) (*n* = 17)	3.4 (0.4) (*n* = 23)	0.994	0.994
IL-3 (pg/mL)	8.9 (1.3) (*n* = 20)	7.8 (1.4) (*n* = 17)	9.7 (1.1) (*n* = 25)	0.576	0.644
IL-4 (pg/mL)	24.2 (5.5) (*n* = 15)	18.1 (5.4) (*n* = 16)	25.9 (4.6) (*n* = 22)	0.556	0.494
IL-5 (pg/mL)	3.2 (0.6) (*n* = 20)	4.1 (0.7) (*n* = 15)	2.9 (0.5) (*n* = 25)	0.451	0.492
IL-6 (pg/mL)	10.9 (5.4) (*n* = 16)	22.4 (6.6) (*n* = 10)	18.2 (5.6) (*n* = 15)	0.385	0.544
IL-7 (pg/mL)	20.1 (1.9) (*n* = 21)	18.5 (2.2) (*n* = 18)	21.5 (1.8) (*n* = 26)	0.591	0.658
IL-8 (pg/mL)	7.2 (1.6) (*n* = 22)	9.1 (1.8) (*n* = 18)	7.4 (1.5) (*n* = 25)	0.708	0.812
IP-10 (pg/mL)	519.8 (48.6) (*n* = 22)	552.9 (55.3) (*n* = 18)	528.2 (45.8) (*n* = 26)	0.901	0.967
MCP-1 (pg/mL)	379.4 (25.9) (*n* = 22)	305.7 (29.4) (*n* = 18)	317.3 (24.3) (*n* = 26)	0.116	0.142
MIP-1α (pg/mL)	4.3 (0.9) (*n* = 10) ^a^	11.7 (1.2) (*n* = 6) ^a,b^	6.9 (1.0) (*n* = 9) ^b^	<0.001	0.001
MIP-1β (pg/mL)	32.5 (1.9) (*n* = 22)	28.6 (2.2) (*n* = 18)	29.0 (1.8) (*n* = 26)	0.324	0.350
TNF-α (pg/mL)	24.8 (1.6) (*n* = 22)	21.8 (1.8) (*n* = 18)	24.6 (1.5) (*n* = 26)	0.387	0.392
TNF-β (pg/mL)	32.1 (15.2) (*n* = 17)	40.2 (17.2) (*n* = 13)	40.4 (14.9) (*n* = 18)	0.913	0.941
VEGF (pg/mL)	88.6 (8.3) (*n* = 22)	95.4 (9.3) (*n* = 18)	88.3 (7.8) (*n* = 25)	0.825	0.799

^a^ Model adjusted for moderate physical activity (METS/week) and following a gluten free diet for at least 18 months. ^b^ Model additionally adjusted for the Mediterranean diet adherence. ^a,b^ superscripts in the same row indicate a significant difference (*p* < 0.05) between groups with the same letter. Values shown as mean (standard error). SOD1 superoxide dismutase 1; TAS, total antioxidant status; IL, interleukin; IP-10, Interferon-inducible protein; IFN-γ, Interferon-γ; MCP-1, Monocyte chemoattractant protein-1; METS, Metabolic equivalents; MIP, Macrophage inflammatory protein; TNF, Tumour necrosis factor; VEGF, Vascular endothelial growth factor.

## Data Availability

The data presented in this study are available on request from the corresponding author. The data are not publicly available due to privacy reasons.

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
