# Peer review of "Influence of Ultra-Processed Foods Consumption on Redox Status and Inflammatory Signaling in Young Celiac Patients"

_nutrients, 2021, doi:10.3390/nu13010156_

Round 1

Reviewer 1 Report

The research presented in the reviewed manuscript “Influence of ultraprocessed-foods consumption on redox status and inflammatory signaling in young celiac patients” is interesting, tooks up the current and important topic. It describes a well-planned and realized experiment. The work concerns not only the adherence to the GF diet, but also, and this is more important, the nutritional value and the quality od products consumed within this therapeutic diet. Although there is an intense search for alternative CD treatments, but the GFD remains  the only effective treatment. Unfortunately, it is not an ideal diet and often it is not properly balanced and contains too many ultra-processed products, which adversely affects the CD subjects health, especially of young patients.

The obtained results have good scientific value. The manuscript is correctly written, the methodology is well described and the results correspond to the conclusions.

Was the study registered at the clinical trials database? if yes, please include this information as well as the registration number both in the methodological part and in the abstract.

And I have a suggestion concerning the future studies - the experimental groups could be similar in numbers, and it may be worth paying attention to a more precise selection of subjects to the control group – a matched case-control could be a suggested model, because in the current study, female predominate in the CD group, although in well-known that in the case of CD, female are twice more often diagnosed then male.

The manuscript may be accepted for publication in its current form

Author Response

ATT:

Alwa Wu

Editor Assistant

Nutrients
---------------------------------------------

December 23th 2020

Dear Editor,

Please, find enclosed the revised version of our manuscript ref1038116 to be considered for publication in Nutrients.

We would like to gratefully thank the Reviewers for their thoughtful and constructive comments, which have undoubtedly improved the quality of our manuscript. We have carefully considered all of the suggestions, and have integrated them into the revised manuscript. Changes to the original manuscript have been incorporated by using yellow background. Moreover, English grammar has been carefully revised by a native (now mentioned in the Acknowledgments section).  We want also to indicate that the correct order of authors and affiliations is the one showed in the own main manuscript file (we made a mistake when introducing them during the submission process).

We believe our manuscript is now stronger as a result of these modifications. An itemized point-by-point response to the Reviewers´ comments is presented below.

REVIEWER 1

Comment 1:

The research presented in the reviewed manuscript “Influence of ultraprocessed-foods consumption on redox status and inflammatory signaling in young celiac patients” is interesting, tooks up the current and important topic. It describes a well-planned and realized experiment. The work concerns not only the adherence to the GF diet, but also, and this is more important, the nutritional value and the quality of products consumed within this therapeutic diet. Although there is an intense search for alternative CD treatments, but the GFD remains the only effective treatment. Unfortunately, it is not an ideal diet and often it is not properly balanced and contains too many ultra-processed products, which adversely affects the CD subjects health, especially of young patients. The obtained results have good scientific value. The manuscript is correctly written, the methodology is well described and the results correspond to the conclusions.

Answer 1:

First of all, thank you for the effort and time devoted to our manuscript as well as for the positive and constructive comments.

Comment 2:

Was the study registered at the clinical trials database? if yes, please include this information as well as the registration number both in the methodological part and in the abstract.

Answer 2:

The reviewer is fully right regarding the recommendation to have the study registered. Unfortunately, due to its cross-sectional design (no intervention was performed) we did not register the present study in any clinical trials database. Notwithstanding, we take note for future studies.

Comment 3:

And I have a suggestion concerning the future studies - the experimental groups could be similar in numbers, and it may be worth paying attention to a more precise selection of subjects to the control group – a matched case-control could be a suggested model, because in the current study, female predominate in the CD group, although in well-known that in the case of CD, female are twice more often diagnosed then male.

Answer 3:

Thanks. We highly appreciate the reviewer suggestion. The recruitment of young aged-matched healthy controls is always a complicated feat.

REVIEWER 2

Comment 1:

This is an interesting work on the effect of excessive consumption of ultraprocessed products (UPF) in the celiac collective. These products have a negative nutritional profile, and the effects of their consumption on this population are unknown. Although the article is interesting and presents new data, some aspects should be clarified.

Answer:

We would like to thank to Reviewer 2 the effort and time devoted to our manuscript. Thank you for the constructive comments that obviously have improved our manuscript.

Major comments:

Comment 2:

The work presents certain weaknesses that have not been declared. On the one hand, the control data are not complete, since neither oxidative/antioxidant biomarkers nor inflammatory markers have been analyzed, which are of great interest to determine whether the oxidative state of celiacs worsens differently depending on the consumption of ultraprocessed foods with respect to healthy individuals.

Answer 2:

Thanks for the comment. Differences between oxidative/antioxidant biomarkers and inflammatory markers have been previously analysed by our group (Diaz-Castro J, Muriel-Neyra C, Martin-Masot R, Moreno J, Maldonado J, Nestares T. Oxidative stress, DNA stability and evoked inflammatory signaling in young celiac patients consuming a gluten-free diet. Eur J Nutr 2020, 59(4), 1577-1584. Doi 10.1007/s00394-019-02013-5) in this study sample, and no clear differences were observed. However, following the reviewer’s advice, differences in oxidative/antioxidant biomarkers and inflammatory profiles in celiac children by percentage of energy intake from ultra-processed foods (below 50% vs. above 50%) and control children are now shown in Table 3. Post-hoc multiple comparisons (with Bonferroni’s correction) has been now applied to examine pair-wise differences between groups (e.g. celiac disease with daily energy intake from ultra-processed foods below 50% vs. controls). This new information has been incorporated along the Manuscript.

Comment 3:

In addition, although the collection of data from 24-hour reminders may be considered adequate, it would be of interest to present data from Food Frequency Questionnaires, which would give us an idea of the general consumption of this group of foods to complement the reminder data.

Answer 3:

We have previously assessed and published (Nestares T, Martín-Masot R, Labella A, Aparicio V, Flor-Alemany M, López-Frías M & Maldonado J. 2020. Is a gluten-free diet enough to maintain a correct iron metabolism in young patients with celiac disease? Nutrients 21;12(3):844. DOI 10.3390/nu12030844) the differences in dietary habits from Food Frequency Questionnaires between celiac and control children. We only found that the consumption of meat derivates was higher in celiac children following a GFD for at least 6 months compared to controls and celiac children following a GFD for less than 6 months. Following the reviewer advice, differences between dietary habits of celiac children by daily intake of ultra-processed foods (below 50% of daily energy intake vs. above 50% of daily energy intake) has been incorporated as Supplementary table 1.

Comment 4:

Finally, the choice of control is not random, since they are children who at some point have consulted a doctor for intestinal problems, and could be a cohort of individuals with a tendency to this type of problem, which could well be due to poor diet.

Answer:

Functional chronic constipation is very common in children, with a prevalence of around 15% of the population and up to 25% of paediatric gastroenterology consultations. The pathophysiology is multifactorial and in more than 95% there is no underlying organic pathology, so we believe that it is a representative group of the general population. Furthermore, as indicated in the exclusion criteria, pathologies with oxidative status or that had dietary alterations in the management of the disease, beyond healthy lifestyle habits (as recommended in patients with functional constipation) were excluded.

References:

- Allen P, Setya A, Lawrence VN. Pediatric Functional Constipation. In: StatPearls. Treasure Island (FL): StatPearls Publishing; November 20, 2020.

- Mugie SM, Benninga MA, Di Lorenzo C. Epidemiology of constipation in children and adults: a systematic review. Best Pract Res Clin Gastroenterol 2011;25:3–18.

Comment 5:

It is obvious that a higher rate of adherence to the Mediterranean diet means less consumption of ultraprocessed foods, by definition. However, although it seems that the data of those celiac who have been on the gluten-free diet for more than 18 months are improving in terms of the consumption of these harmful products, the index of adherence to the Mediterranean diet does not seem to be improving. What can cause this phenomenon? It should be discussed.

Answer:

Although we have observed that the intake of ultraprocessed foods is lower among celiac children with higher Mediterranean diet adherence (as assessed trough the KIDMED score), small differences were found regarding dietary habits. In fact, celiac children with more than 50% of daily energy intake from ultraprocessed foods showed a lower intake of whole dairy products (p=0.006) and a trend to higher intake of poultry (p=0.076) and a lower intake of vegetables (p=0.077). As a result, although a tendency towards healthier habits has been observed in CD children with less than 50% of daily energy intake from ultraprocessed foods, the differences might not be strong enough to reflect changes in the overall dietary pattern. Consequently, the global index of adherence to the MD might not be sensitive enough. This new information has been added in Supplementary table 1 and in the Methods (page 5, lines 235-236), Results (page 9, lines 309-313) and Discussion (page 10, lines 400-410) sections.

Comment 6:

Although those individuals with a higher consumption of UPF show greater adherence to MD according to Figure 1, no data adjusted for that adherence to MD are shown in the case of the biomarker data.  Moreover, this fact is discussed in the last line of the discussion, where it is argued that other authors have not found a relationship between increased incidence of respiratory disease and adherence to MD. In line 371, for its part, the authors state that they have seen a more favorable redox profile in those individuals with greater adherence, but in reality it should be said that they have seen it in those who consumed less than 50% of the energy in the form of UPF. The reviewer believes that it would be important to corroborate this relationship directly (adherence to MD vs. redox/anti-inflammatory profile), given that they have the data.

Answer:

The reviewer is fully right. The sentence “In addition, patients with more adherence to the MD presented a more favorable redox and pro-inflammatory profile than those with more UPF consumption and less adherence to MD” has been replaced by "In addition, patients who consumed less than 50% of their energy intake from UPF presented a more favourable oxidative-antioxidant and inflammatory profiles than those with greater UPF consumption". We further explored the association between MD adherence and redox/anti-inflammatory profile among CD children finding no significant association (data not shown). In fact, even after adjusting our results for MD adherence, patients who consumed less than 50% of their daily energy intake from UPF presented a more favorable oxidative-antioxidant and inflammatory profiles than those with greater UPF consumption. Moreover, a new Table 3 assessing differences in oxidative/antioxidant biomarkers and inflammatory profiles in CD children by percentage of energy intake from ultraprocessed foods (below 50% vs. above 50%) has been added.

Comment 7:

It is necessary to clarify whether they introduced the specific composition of gluten-free UPFs (from labels) when calculating dietary balance, since there is a lot of literature that speaks of the different composition of foods with gluten and those specifically formulated without gluten. This fact may influence the caloric contribution, especially in those whose % of energy consumed from these foods is so high. Although it has probably been done this way, it is not specified in the materials and methods. If not, it is a clear weakness.

Answer:

Thanks for catching this. The specific composition of gluten and gluten-free UPFs (from labels) were included in the Evalfinut software when calculating the dietary balance. This information has been now detailed into the Methods section (page 5, lines 203-204).

Minor comments:

Comment 8: Differences or LEVELS???.

Answer:  Dear reviewer, we do not understand to what you specifically refer. Please, give us more details.

Comment 9: Substitutes natural gluten CONTAINING products…

Answer: Thanks. Corrected.

Comment 10: Table 1. Parents´ marital status, in the case of control group, % symbol should be deleted.

Answer: Thanks. Amended.

Reviewer 2 Report

This is an interesting work on the effect of excessive consumption of ultraprocessed products (UPF) in the celiac collective. These products have a negative nutritional profile, and the effects of their consumption on this population are unknown.

Although the article is interesting and presents new data, some aspects should be clarified.

Major comments:

- The work presents certain weaknesses that have not been declared. On the one hand, the control data are not complete, since neither oxidative/antioxidant biomarkers nor inflammatory markers have been analyzed, which are of great interest to determine whether the oxidative state of celiacs worsens differently depending on the consumption of ultraprocessed foods with respect to healthy individuals. In addition, although the collection of data from 24-hour reminders may be considered adequate, it would be of interest to present data from Food Frequency Questionnaires, which would give us an idea of the general consumption of this group of foods to complement the reminder data. Finally, the choice of control is not random, since they are children who at some point have consulted a doctor for intestinal problems, and could be a cohort of individuals with a tendency to this type of problem, which could well be due to poor diet.

- It is obvious that a higher rate of adherence to the Mediterranean diet means less consumption of ultraprocessed foods, by definition. However, although it seems that the data of those coeliacs who have been on the gluten-free diet for more than 18 months are improving in terms of the consumption of these harmful products, the index of adherence to the Mediterranean diet does not seem to be improving. What can cause this phenomenon? It should be discussed.

- 50% of energy in the form of ultraprocessed products seems a very high figure. In fact, although the control data is below the coeliac group´s one, it is also quite high. I consider of great interest an analysis of those products that these patients consume in substitution of gluten containing products and compare them with those that both coeliacs and non-celiacs consume. That is, what UPF foods are contributing to a higher consumption of these products in celiacs compared to healthy individuals.

- Although those individuals with a higher consumption of UPF show greater adherence to MD according to Figure 1, no data adjusted for that adherence to MD are shown in the case of the biomarker data.  Moreover, this fact is discussed in the last line of the discussion, where it is argued that other authors have not found a relationship between increased incidence of respiratory disease and adherence to MD. In line 371, for its part, the authors state that they have seen a more favorable redox profile in those individuals with greater adherence, but in reality it should be said that they have seen it in those who consumed less than 50% of the energy in the form of UPF. The reviewer believes that it would be important to corroborate this relationship directly (adherence to MD vs. redox/anti-inflammatory profile), given that they have the data.

- It is necessary to clarify whether they introduced the specific composition of gluten-free UPFs (from labels) when calculating dietary balance, since there is a lot of literature that speaks of the different composition of foods with gluten and those specifically formulated without gluten. This fact may influence the caloric contribution, especially in those whose % of energy consumed from these foods is so high. Although it has probably been done this way, it is not specified in the materials and methods. If not, it is a clear weakness.

Minor comments:

  • Differences or LEVELS???

  • Substitutes natural gluten CONTAINING products…

  • Table 1. Parents´marital status, in the case of control group, % symbol should be deleted.

Author Response

REVIEWER 2

Comment 1:

This is an interesting work on the effect of excessive consumption of ultraprocessed products (UPF) in the celiac collective. These products have a negative nutritional profile, and the effects of their consumption on this population are unknown. Although the article is interesting and presents new data, some aspects should be clarified.

Answer:

We would like to thank to Reviewer 2 the effort and time devoted to our manuscript. Thank you for the constructive comments that obviously have improved our manuscript.

Major comments:

Comment 2:

The work presents certain weaknesses that have not been declared. On the one hand, the control data are not complete, since neither oxidative/antioxidant biomarkers nor inflammatory markers have been analyzed, which are of great interest to determine whether the oxidative state of celiacs worsens differently depending on the consumption of ultraprocessed foods with respect to healthy individuals.

Answer 2:

Thanks for the comment. Differences between oxidative/antioxidant biomarkers and inflammatory markers have been previously analysed by our group (Diaz-Castro J, Muriel-Neyra C, Martin-Masot R, Moreno J, Maldonado J, Nestares T. Oxidative stress, DNA stability and evoked inflammatory signaling in young celiac patients consuming a gluten-free diet. Eur J Nutr 2020, 59(4), 1577-1584. Doi 10.1007/s00394-019-02013-5) in this study sample, and no clear differences were observed. However, following the reviewer’s advice, differences in oxidative/antioxidant biomarkers and inflammatory profiles in celiac children by percentage of energy intake from ultra-processed foods (below 50% vs. above 50%) and control children are now shown in Table 3. Post-hoc multiple comparisons (with Bonferroni’s correction) has been now applied to examine pair-wise differences between groups (e.g. celiac disease with daily energy intake from ultra-processed foods below 50% vs. controls). This new information has been incorporated along the Manuscript.

Comment 3:

In addition, although the collection of data from 24-hour reminders may be considered adequate, it would be of interest to present data from Food Frequency Questionnaires, which would give us an idea of the general consumption of this group of foods to complement the reminder data.

Answer 3:

We have previously assessed and published (Nestares T, Martín-Masot R, Labella A, Aparicio V, Flor-Alemany M, López-Frías M & Maldonado J. 2020. Is a gluten-free diet enough to maintain a correct iron metabolism in young patients with celiac disease? Nutrients 21;12(3):844. DOI 10.3390/nu12030844) the differences in dietary habits from Food Frequency Questionnaires between celiac and control children. We only found that the consumption of meat derivates was higher in celiac children following a GFD for at least 6 months compared to controls and celiac children following a GFD for less than 6 months. Following the reviewer advice, differences between dietary habits of celiac children by daily intake of ultra-processed foods (below 50% of daily energy intake vs. above 50% of daily energy intake) has been incorporated as Supplementary table 1.

Comment 4:

Finally, the choice of control is not random, since they are children who at some point have consulted a doctor for intestinal problems, and could be a cohort of individuals with a tendency to this type of problem, which could well be due to poor diet.

Answer:

Functional chronic constipation is very common in children, with a prevalence of around 15% of the population and up to 25% of paediatric gastroenterology consultations. The pathophysiology is multifactorial and in more than 95% there is no underlying organic pathology, so we believe that it is a representative group of the general population. Furthermore, as indicated in the exclusion criteria, pathologies with oxidative status or that had dietary alterations in the management of the disease, beyond healthy lifestyle habits (as recommended in patients with functional constipation) were excluded.

References:

- Allen P, Setya A, Lawrence VN. Pediatric Functional Constipation. In: StatPearls. Treasure Island (FL): StatPearls Publishing; November 20, 2020.

- Mugie SM, Benninga MA, Di Lorenzo C. Epidemiology of constipation in children and adults: a systematic review. Best Pract Res Clin Gastroenterol 2011;25:3–18.

Comment 5:

It is obvious that a higher rate of adherence to the Mediterranean diet means less consumption of ultraprocessed foods, by definition. However, although it seems that the data of those celiac who have been on the gluten-free diet for more than 18 months are improving in terms of the consumption of these harmful products, the index of adherence to the Mediterranean diet does not seem to be improving. What can cause this phenomenon? It should be discussed.

Answer:

Although we have observed that the intake of ultraprocessed foods is lower among celiac children with higher Mediterranean diet adherence (as assessed trough the KIDMED score), small differences were found regarding dietary habits. In fact, celiac children with more than 50% of daily energy intake from ultraprocessed foods showed a lower intake of whole dairy products (p=0.006) and a trend to higher intake of poultry (p=0.076) and a lower intake of vegetables (p=0.077). As a result, although a tendency towards healthier habits has been observed in CD children with less than 50% of daily energy intake from ultraprocessed foods, the differences might not be strong enough to reflect changes in the overall dietary pattern. Consequently, the global index of adherence to the MD might not be sensitive enough. This new information has been added in Supplementary table 1 and in the Methods (page 5, lines 235-236), Results (page 9, lines 309-313) and Discussion (page 10, lines 400-410) sections.

Comment 6:

Although those individuals with a higher consumption of UPF show greater adherence to MD according to Figure 1, no data adjusted for that adherence to MD are shown in the case of the biomarker data.  Moreover, this fact is discussed in the last line of the discussion, where it is argued that other authors have not found a relationship between increased incidence of respiratory disease and adherence to MD. In line 371, for its part, the authors state that they have seen a more favorable redox profile in those individuals with greater adherence, but in reality it should be said that they have seen it in those who consumed less than 50% of the energy in the form of UPF. The reviewer believes that it would be important to corroborate this relationship directly (adherence to MD vs. redox/anti-inflammatory profile), given that they have the data.

Answer:

The reviewer is fully right. The sentence “In addition, patients with more adherence to the MD presented a more favorable redox and pro-inflammatory profile than those with more UPF consumption and less adherence to MD” has been replaced by "In addition, patients who consumed less than 50% of their energy intake from UPF presented a more favourable oxidative-antioxidant and inflammatory profiles than those with greater UPF consumption". We further explored the association between MD adherence and redox/anti-inflammatory profile among CD children finding no significant association (data not shown). In fact, even after adjusting our results for MD adherence, patients who consumed less than 50% of their daily energy intake from UPF presented a more favorable oxidative-antioxidant and inflammatory profiles than those with greater UPF consumption. Moreover, a new Table 3 assessing differences in oxidative/antioxidant biomarkers and inflammatory profiles in CD children by percentage of energy intake from ultraprocessed foods (below 50% vs. above 50%) has been added.

Comment 7:

It is necessary to clarify whether they introduced the specific composition of gluten-free UPFs (from labels) when calculating dietary balance, since there is a lot of literature that speaks of the different composition of foods with gluten and those specifically formulated without gluten. This fact may influence the caloric contribution, especially in those whose % of energy consumed from these foods is so high. Although it has probably been done this way, it is not specified in the materials and methods. If not, it is a clear weakness.

Answer:

Thanks for catching this. The specific composition of gluten and gluten-free UPFs (from labels) were included in the Evalfinut software when calculating the dietary balance. This information has been now detailed into the Methods section (page 5, lines 203-204).

Minor comments:

Comment 8: Differences or LEVELS???.

Answer:  Dear reviewer, we do not understand to what you specifically refer. Please, give us more details.

Comment 9: Substitutes natural gluten CONTAINING products…

Answer: Thanks. Corrected.

Comment 10: Table 1. Parents´ marital status, in the case of control group, % symbol should be deleted.

Answer: Thanks. Amended.

Round 2

Reviewer 2 Report

Dear authors,

I am very pleased with the review provided, and although I still have minor comments, I must congratulate you on your very interesting article.

Minor comments:

-Table 3.

  • In the case of 15-F2t-isoprostanes, isn't the letter corresponding to the statistics missing in the control group?
  • In the control group, in the value corresponding to the standard deviation of SOD1, one decimal is missing.
  • There are several things in this table that have caught my attention. It is surprising that in the case of the IFN-g, the children who consumed less than 50% of their energy from UPF and the children in the control group are statistically different from each other (they have a different superscript letter), but not from the group that consumed more than 50%. Is the effect of the high variability in this parameter really so important in these two groups to show this phenomenon? Something similar occurs with the levels of IPM-1α and SOD1, where groups that at first glance appear to be the same turn out not to be, but they do appear to be statistically equal to an apparently more distant group. Moreover, in the text the information is in line with considering the closest groups as equal, contrary to what the reviewer interprets in the table in view of the letters that show the differences between groups. In addition, it may be necessary to clarify what those letters in the table mean.

- With respect to L361 of the previous manuscript, the reviewer apologizes for not having provided sufficient information for consideration of his commentary. In any case, in the new version the sentence is correct (L390). What was minor was not the differences, but the concentrations or levels of SOD.

Author Response

Comment:

I am very pleased with the review provided, and although I still have minor comments, I must congratulate you on your very interesting article.

Answer:

Thank you again for your constructive review, which has undoubtedly improved the quality of our manuscript.

Comment:

In the case of 15-F2t-isoprostanes, isn't the letter corresponding to the statistics missing in the control group? In the control group, in the value corresponding to the standard deviation of SOD1, one decimal is missing.

Answer:

The reviewer is right. Thanks for catching this. Amended.

Comment:
There are several things in this table that have caught my attention. It is surprising that in the case of the IFN-g, the children who consumed less than 50% of their energy from UPF and the children in the control group are statistically different from each other (they have a different superscript letter), but not from the group that consumed more than 50%. Is the effect of the high variability in this parameter really so important in these two groups to show this phenomenon?

Answer:

Dear reviewer, taken into account your interesting appreciation, we have identified potential outliers and repeated all the statistics (see Table3). Nonetheless, we believe that the main reason why there is a borderline but no significant pairwise difference between some groups is due to the fact that Bonferroni’s adjustment is characterized by being a strict statistical method. In fact, the significant overall P suggest that there are evidence of significations among more groups than those statistically significant. For instance, regarding IFN-g, between celiac children below 50% versus celiac children above 50% of energy intake from ultra-processed foods.

Comment:

Something similar occurs with the levels of IPM-1α and SOD1, where groups that at first glance appear to be the same turn out not to be, but they do appear to be statistically equal to an apparently more distant group. Moreover, in the text the information is in line with considering the closest groups as equal, contrary to what the reviewer interprets in the table in view of the letters that show the differences between groups. In addition, it may be necessary to clarify what those letters in the table mean.

Answer:
Sorry for the forgetfulness. We have now clarified in the table 3 what the same letters mean (i.e. “a, b superscripts in the same row indicate a significant difference (p<0.05) between groups with the same letter) and, as mentioned above, repeated and revised all the pairwise comparisons.